# Glycoproteomic Analysis of Urinary Extracellular Vesicles for Biomarkers of Hepatocellular Carcinoma

**DOI:** 10.3390/molecules28031293

**Published:** 2023-01-29

**Authors:** Dejun Li, Shengnan Jia, Shuyue Wang, Lianghai Hu

**Affiliations:** 1Center for Supramolecular Chemical Biology, School of Life Sciences, Jilin University, Changchun 130012, China; 2Prenatal Diagnosis Center, Reproductive Medicine Center, The First Hospital of Jilin University, Changchun 130021, China; 3Department of Hepatopancreatobiliary Medicine, The Second Hospital, Jilin University, Changchun 130041, China

**Keywords:** hepatocellular carcinoma, glycoproteomic, N-glycosylation, extracellular vesicles, mass spectrometry, biomarker

## Abstract

Hepatocellular carcinoma (HCC) accounts for the most common form of primary liver cancer cases and constitutes a major health problem worldwide. The diagnosis of HCC is still challenging due to the low sensitivity and specificity of the serum α-fetoprotein (AFP) diagnostic method. Extracellular vesicles (EVs) are heterogeneous populations of phospholipid bilayer-enclosed vesicles that can be found in many biological fluids, and have great potential as circulating biomarkers for biomarker discovery and disease diagnosis. Protein glycosylation plays crucial roles in many biological processes and aberrant glycosylation is a hallmark of cancer. Herein, we performed a comprehensive glycoproteomic profiling of urinary EVs at the intact N-glycopeptide level to screen potential biomarkers for the diagnosis of HCC. With the control of the spectrum-level false discovery rate ≤1%, 756 intact N-glycopeptides with 154 N-glycosites, 158 peptide backbones, and 107 N-glycoproteins were identified. Out of 756 intact N-glycopeptides, 344 differentially expressed intact N-glycopeptides (DEGPs) were identified, corresponding to 308 upregulated and 36 downregulated N-glycopeptides, respectively. Compared to normal control (NC), the glycoproteins LG3BP, PIGR and KNG1 are upregulated in HCC-derived EVs, while ASPP2 is downregulated. The findings demonstrated that specific site-specific glycoforms in these glycoproteins from urinary EVs could be potential and efficient non-invasive candidate biomarkers for HCC diagnosis.

## 1. Introduction

Liver cancer is the sixth most commonly diagnosed cancer and the third leading cause of cancer death worldwide in 2020, with approximately 906,000 new cases and 830,000 deaths [1]. It is estimated that, by 2025, more than 1 million individuals will be affected by liver cancer annually [2]. Hepatocellular carcinoma (HCC) accounts for 75–85% of primary liver cancer cases and constitutes a major health problem worldwide [3]. Although patients with HCC can be treated with local ablation, surgical resection, and liver transplantation, due to the lack of specific clinical symptoms and efficient diagnostic method in the early stage, the majority of patients are already in advanced stages when they are diagnosed, thus having poor prognosis. For example, the 5-year survival rate for HCC dropped from 70% for early-stage patients to 16% for late-stage patients [4]. Therefore, early detection of HCC is of particular importance to reduce the mortality rates of HCC patients. Currently, serum α-fetoprotein (AFP) measurement and liver ultrasonography are widely accepted as the most effective and affordable tools to screen and diagnose HCC in the clinic [5]. However, not all patients with HCC have elevated serum AFP, and the sensitivity and specificity of AFP are limited to 40–62% [6]. Furthermore, tumors can only be detected by imaging technology if they are greater than 1 cm in diameter [7]. Therefore, there is an urgent need to develop non-invasive biomarkers that are more sensitive for HCC diagnosis at the early stage.

In recent years, liquid biopsy has gained much attention as an alternative biomarker to conventional biomarkers and tissue biopsy for cancer diagnosis and monitoring. Liquid biopsy represents a minimally invasive procedure that is usually performed on biofluid samples to detect circulating tumor by-products, which includes circulating tumor cells (CTCs), cell-free DNA (cfDNA), and extracellular vesicles (EVs) [8,9,10]. Compared with CTCs and cfDNA, EVs have attracted researchers’ interest because they are not only present in circulation at relatively early stages of disease, but also with high abundance and high stability [11,12]. EVs are heterogeneous populations of phospholipid bilayer-enclosed vesicles that are released from almost all types of cells and can be found in biological fluids such as blood, saliva, urine, breast milk, and cerebrospinal fluid (CSF) [13,14]. According to their biophysical properties and biogenesis pathways, EVs are generally grouped into two subclasses, exosomes and microvesicles. Exosomes are small EVs with a size range of ~30–150 nm that derived from the endosomal membrane pathway, while microvesicles are larger than exosomes (100–1000 nm) and are formed by budding directly from the cell membrane [11]. EVs play a central role in intercellular communication by selectively packaging and transporting bioactive cargoes such as proteins, RNA and metabolites to recipient cells, and participating in immune responses, cardiovascular diseases and cancer [15,16]. Recent studies have found that cancer cell-derived EVs released into the tumor microenvironment and circulation can promote tumor progression and metastasis by inducing matrix remodeling, angiogenesis, inflammation, and metastatic niche formation [17]. Furthermore, multiple studies on EV-based diagnosis of cancer have suggested that EV transcriptomic and proteomic biomarkers can increase the likelihood of early and sensitive detections of cancer. For example, plasma EV-derived miRNA-483-3 could be used as a potential biomarker for the diagnosis of early-stage small cell lung cancer, and miRNA-152-3p and miRNA-1277-5p could be used for the diagnosis of early-stage non-small cell lung cancer [18]. Glypican 1 (GCP1) and macrophage migration inhibitory factor (MIF) were identified as potential exosome-associated biomarkers for early diagnosis of pancreatic cancer [19,20]. As has been reported, tissue- and plasma-derived exosomal hsa-miR-483-5p could differentiate HCC and non-HCC cases and may represents a potential specific and sensitive biomarker for HCC diagnosis [21]. However, while these studies mainly focus on the utilization of proteins or RNAs in EVs as biomarkers for disease diagnosis, little attention is paid to researching the use of post-translational modifications (PTMs) of proteins in EVs as biomarker.

Protein glycosylation is one of the most important PTMs found in eukaryotic cells, which is involved in molecular recognition, immune response, and intercellular communication [22,23,24]. Notably, aberrant glycosylation of proteins is a hallmark of cancer, and glycoproteins account for the majority of FDA-approved biomarkers for cancer diagnosis and monitoring [25]. As has been reported, bifucosylated N-glycans with both core and antennary fucosylation were elevated in HCC patients as compared to patients with cirrhosis [26]. Moreover, AFP-L3, a corefucosylated variant of AFP which reacts with LCA fraction, was known to be more specific to HCC than AFP and was a useful biomarker for HCC diagnosis and assessment [27]. In recent years, studies on EV glycosylation have gradually increased. Because EVs originate from the plasma membrane or from multivesicular endosomes, their surface is highly glycosylated, and specific cancer-associated glycosignatures have already been identified in cancer EVs, thus highlighting EV glycosylation as a potential source of novel circulating biomarkers in cancer diagnosis [28,29]. Currently, studies have been reported on the glycosylation of EVs derived from different tumor cell lines and biofluids. For example, the proteoglycan glypican-1 positive (GPC1^+^) exosomes derived from the serum of pancreas cancer patients could distinguish early- and late-stage pancreas cancer as well as patients with a benign pancreas disease [30]. Furthermore, Sakaue et al. found that increased levels of sialic acids of CD133 glycoprotein from the pancreatic cancer patient’s ascites-derived exosomes, and indicated that highly glycosylated CD133 could be a potential biomarker for advanced pancreatic cancer [31]. Moreover, the proteoglycan biglycan (BGN) and the glycoprotein basigin (EMMPRIN) were found to be enriched in pancreatic tumor tissue-derived EVs [32]. In addition, high levels of EMMPRIN were observed from breast cancer patient serumderived MVs [33]. Notably, galectin-3-binding protein (LG3BP) was identified in several tumor-derived EVs, including ovarian cancer, uveal melanoma, and gliomas [34,35,36]. With respect to HCC, there are currently only two glycosylation studies on EVs that have been reported, one from serum-derived exosome, and the other from cell line-derived exosome. Lv et al. performed a reverse capture strategy to profile the N-glycome of exosomes isolated from HCC patients’ serum samples, and compared this to the N-glycans identified in exosomes from healthy samples. The majority of the N-glycans detected in exosomes from HCC patients were modified with sialic acids or fucoses [37]. Wang et al. found that compared to exosomes derived from naive HCC cells, α2,6-sialylation degradation abolished both the proliferation-promoting and migration-promoting effects of HCC-exo, which suggests that a loss of α2,6-sialylation decreases HCC progression through the loss of cancer cell-derived exosomes [38]. The above studies indicate that, compared with normal EVs, specific glycosylation modifications are existing on the surface of tumor cell-derived EVs, therefore they can be used as potential biomarkers for disease diagnosis and prognosis. Nevertheless, previously studies on glycosylation of EVs mainly focus on the identification of released glycans or deglycosylated peptides, losing the site-specific information of each individual glycan. Little is known about the information on the glycosylation of tumor-derived EVs at the intact glycopeptide (including both the glycosylation sites and glycan structures) level. 

In this study, we investigated the glycoproteomic profiling of EVs from HCC patient urine samples to screen the potential candidate biomarker. Urinary EVs were first isolated by a previously reported EVTRAP method [39]. Then, intact-glycoproteomic analysis of EVs glycoproteins were detected using the C18-HILIC-MS/MS analysis with the N-glycan database engine GPSeeker search [40]. In total, 756 intact N-glycopeptides mapping to 154 N-glycosites on 107 intact N-glycoproteins were identified. Furthermore, variations in the N-glycans and site-specific glycans were elucidated. With the criteria of differentially expressed intact N-glycopeptides (DEGPs) as fold change cutoff ratio ≥2.0 or ≤0.5, 344 DEGPs were identified from EVs in HCC relative to the NC group, corresponding to 308 upregulated and 36 downregulated N-glycopeptides, respectively. Moreover, the relationship between aberrated glycoproteins in EVs and HCC was further discussed.

## 2. Results and Discussion

### 2.1. Workflow of EVs Proteomics and Glycoproteomics from Urine Samples

The workflow of EVs proteomics and glycoproteomics analysis from urine samples is shown in Figure 1. In brief, EVs were first isolated by EVTRAP beads using pooled urine samples from HCC patients and from NC, respectively. Then, urinary EVs were lysed and the extracted proteins were digested into peptides; thereafter, 5% of the digested peptides were directly used to perform proteomic analysis with nanoflow LC-MS/MS. The other 95% portion of peptides were labeled with isotopomeric dimethyl labels and mixed at a 1:1 ratio from HCC and NC, and the N-glycopeptides enrichment was performed by the HILIC method. Finally, the N-glycopeptides were analyzed by nanoflow LC-MS/MS and data were analyzed with the GPSeeker search engine.

### 2.2. General Description of Extracellular Vesicles Isolated by EVTRAP

Compared to the ultracentrifugation method, the EVTRAP method had a high recovery yield (over 95%) from urine samples [39], and had the capacity to uncover more low-abundant urinary biomarkers, especially EV proteins with PTMs such as phosphorylation and glycosylation. The quality of the isolated EVs was evaluated by proteins identified in the proteomic analysis with mass spectrometry. In three replicate tests, 1331 and 1275 proteins were identified in urinary EVs from the HCC and NC group, respectively (Appendix A). The quality of the isolated EVs was evaluated by comparing the proteins identified in EVs by direct mass spectrometry with that in the Vesiclepedia database [41], and up to 96% of the proteins were matched form the HCC group (Figure 2a). Moreover, 94 proteins were overlapped with the top 100 most common identified EV marker proteins, including EV markers such as CD9, CD63, Alix, TSG101, annexin A1 and syntenin-1 (Figure 2b). In addition, Gene Ontology (GO) analysis of cellular component revealed “extracellular exosome” as the top enriched GO item (Figure 2c). The quality evaluation results of EVs from the NC group was shown in Appendix A. Therefore, after quality assessment of the isolated EVs, we performed comprehensive glycoproteomic analysis of the EVs at the intact N-glycopeptide level and evaluated its potential role as a biomarker for the hepatocellular carcinoma.

### 2.3. Comprehensive Analysis of Intact N-glycopeptides in the Urinary EVs

With target and decoy database searches using the intact N-glycopeptide search engine GPSeeker, a total of 756 intact N-glycopeptides were identified, corresponding to 154 N-glycosites on 107 intact N-glycoproteins with 286 putative N-glycan linkages from 94 monosaccharide compositions (Figure 3a, Appendix A). All of the identified N-glycosites were observed with N-X-S and N-X-T sequons, in which ~60% of the glycosites had the N-X-T sequons and 40% had N-X-S sequons (Figure 3b). In addition, the percentage of mannose, hybrid, and complex N-glycosylation among the 756 intact N-glycopeptide IDs is 30.3%, 14.9% and 54.8%, respectively. Furthermore, compared with the NC-derived EVs, the percentage of mannose N-glycosylation in HCC-derived EVs was increased while the hybrid and complex N-glycosylation showed the opposite trend (Figure 3c). Moreover, for the monosaccharide compositions of the N-glycan moieties among the identified intact N-glycopeptides, the composition containing two HexNAc was the most common, and the eight most common monosaccharide compositions and their occurrence numbers are shown in Appendix A. Macro-heterogeneity and micro-heterogeneity are two key features of protein glycosylation; therefore, we further performed the heterogeneity of glycosylation analysis with our data set. As for macro-heterogeneity, approximately 30% of the glycoproteins were identified to have more than one N-glycosite; whereas for micro-heterogeneity, ~46% of the glycosites were occupied by more than one N-linked glycan compositions (Figure 3d). For example, on probable serine carboxypeptidase (Q9H3G5, CPVL_HUMAN), the same N-glycan composition N2H5F0S0 was identified on three N-glycosites (N81, N307, and N346); while on the same N-glycosite N346, other N-glycan compositions (N2H3F0S0, N2H4F1S0 and N2H4F0S0) were also identified (Table 1).

### 2.4. Relatively Quantification Analysis of the Site-Specific Glycans in the Urinary EVs

To further analyze the site-specific glycans in urinary EVs, the abundance of the 756 intact N-glycopeptide IDs together with their isotopic pairs (4.0134 Da) in the corresponding MS spectra were searched with GPSeekerQuan. With the criteria of observation of all of the six most abundant isotopic peaks; and with the designated differentially expressed intact N-glycopeptides (DEGPs) as fold change cutoff ratio ≥2.0 or ≤0.5, a total of 344 DEGPs were identified corresponding to 308 upregulated and 36 downregulated N-glycopeptides, respectively (Appendix A). The 308 upregulated intact N-glycopeptides come from 51 N-glycoproteins and the 36 downregulated intact N-glycopeptides come from 7 N-glycoproteins. For the 51 intact N-glycoproteins with up-regulation, 38 were quantified with more than one DEGPs. Among the upregulated N-glycopeptides, intact N-glycopeptide NVTLLSR_N2H9F0S0 from N-glycosite N828 of N-glycoprotein Glutamyl aminopeptidase (AMPE_HUMAN, Q07075) displayed the highest up-regulation, with a fold change of 40 in HCC relative to NC (Appendix A). For Galectin-3-binding protein (LG3BP_HUMAN, Q08380), on N-glycosite N398, intact N-glycopeptide GLNLTEDTYKPR_N2H3F1S0 was found to be 9.31-fold upregulated in HCC relative to NC (Figure 4), while on N-glycosite N551, intact N-glycopeptides AAIPSALDTNSSK with compositions of N2H3F1S0, N2H5F0S0 and N6H6F1S0 were identified with a range of upregulated (2.21–9.31). LG3BP is a large oligomeric glycoprotein present in human body fluids and originally identified as a tumor-secreted antigen from breast and lung cancer [42,43]. Upregulation of LG3BP was previously observed in the serum of patients with HCC, and high expression of LG3BP was as a poor prognostic biomarker for HCC [44,45]. Recently, proteomic analysis on serum exosomes revealed that the LG3BP has higher diagnostic capacity than AFP, and has the potential as a candidate biomarker for HCC detection [46,47]. In addition, upregulation of intact N-glycopeptideWNNTGCQALPSQDEGPSK_N2H5F0S0 at N-glycosite N499 of the polymeric immunoglobulin receptor (PIGR_HUMAN, P01833) was quantified with a fold change of 2.25 in HCC relative to NC (Appendix A). PIGR is universally expressed in epithelial cells and regulates transcytosis of dimeric IgA and pentameric IgM, which are the first-line antibodies in response to initial infection. Previously, studies have shown that PIGR was significantly overexpressed in tumor tissues and in serum samples in HCC patients, and the level of PIGR was statistically significantly associated with early recurrence in early-stage HCC and in hepatitis B surface antigen-positive HCC patients [48,49]. Previous studies have shown that elevated PIGR levels were found in the serum EVs from patients with HCC, and EV-PIGR could promote liver cancer cell aggressiveness through activating thePDK1/Akt/GSK3β/β-catenin signaling axis [46,50]. Therefore, EV-PIGR holds the potential as a diagnostic and prognostic marker in HCC. For Kininogen-1 (KNG1_HUMAN, P01042), intact N-glycopeptides with compositions of N6H3F0S0, N5H4F1S1 and N6H3F2S0 were identified with a range of upregulated (2.27–3.39) in HCC relative to NC. KNG1 is a precursor for kinins and plays a central role in the blood coagulation system and in the kinin–kallikrein system [51]. Previous studies have shown that the serum level of KNG1 was significantly elevated in patients with various cancers, including advanced colorectal adenoma (ACA), colorectal cancer (CRC), and HCC [52,53]. Hence, KNG1 is a potential serum biomarker for the early detection of ACA, CRC and HCC. Moreover, we found that the N-glycan of KNG1 possesses fucose modification, and this is consistent with the results from Wang et al. [54], who also revealed an increased amount of both core and outer-arm fucosylation of KNG1 in patients with HCC. In addition, they revealed that the increased reactivity with fucose-binding lectin observed in patients with HCC is the result of increased levels of fucosylation. Therefore, they proposed a biomarker algorithm that use the combination of fucosylated KNG1, AFP, and clinical characteristics to improve the detection of early-stage HCC. 

For the seven intact N-glycoproteins with down-regulation, all of them were quantified with more than one DEGP. Among the downregulated N-glycopeptides, intact N-glycopeptide SVNVSLTQEELDSGLDELSVR_N4H5F0S0 from N-glycosite N463 of A-kinase anchor protein 2 (AKAP2_HUMAN, Q9Y2D5) displayed the lowest down-regulation, with the fold change of 0.18 in HCC relative to NC (Appendix A). Intact N-glycopeptideKNQSSEDILR_N4H5F1S1 at N-glycosite N478 of apoptosis stimulating of p53 protein 2 (ASPP2_HUMAN, Q13625) was downregulated with a fold change of 0.34 in HCC relative to NC (Figure 5). ASPP2, a member of the ankyrin-repeat, SH3-domain, and proline-rich region containing protein (ASPP) family, which also include ASPP1 and inhibitory ASPP (iASPP), is a key regulator of various cancer-relevant cellular phenotypes and behaviors, including cell proliferation, apoptosis, and polarity [55,56]. ASPP2 is a haplo-insufficient tumor suppressor, and is frequently downregulated in a variety of human cancers, including breast cancer, lung cancer and leukemia [57]. Furthermore, a previous study by Zhao et al. [58] also found that ASPP2 is downregulated in hepatocellular carcinoma owing to DNA methylation. Chen et al. [59] revealed that downregulation of ASPP2 may contribute to tumor progression and chemoresistance via promoting BECN1-dependent autophagy in HCC. Moreover, studies have shown that reduced ASPP2 expression results in EMT, and is associated with poor survival in hepatocellular carcinoma patients [60]. Therefore, ASPP2 plays important roles in the development of HCC, which may be a potential target for the treatment of HCC.

Interestingly, there were 5 glycoproteins with both upregulated and downregulated site-specific glycans. For example, on Alpha-2-macroglobulin-like protein 1 (A2ML1_HUMAN, A8K2U0), intact N-glycopeptide LGHI^867^NFTISTK_N2H5F0S0 was quantified with up-regulation while THHW^857^NITAVK_N2H8F0S0 was significantly downregulated. Furthermore, there were N-glycosites with both upregulated and downregulated glycans. For example, on glycoprotein Vasorin (VASN_HUMAN, Q6EMK4), on the same glycosylation site N500, the upregulated intact N-glycopeptide^500^NLSGPDK_N2H5F0S0 and ^500^NLSGPDK_N3H6F0S0 as well as downregulated ^500^NLSGPDK_N3H5F0S1 were observed (Figure 6). VASN was found to be highly expressed in the cells and tissues of HCC, and at a high level in the serum samples of HCC patients [61]. Being a transmembrane glycoprotein, VASN regulates cellular responses to vascular lesion, and exosomal VASN from HepG2 cells could promote the migration of human umbilical vein endothelial cells (HUVEC) [62]. These results suggest that VASN may play an important role in the HCC pathogenesis.

### 2.5. Gene Ontology Analysis of the Differentianl Expressed Intact N-glycopeptides

To further investigate the biological information of the differentially expressed N-glycoproteins identified in the urinary EVs, GO analysis was carried out using DAVID Bioinformatics Resources 6.8 for the differentially expressed intact N-glycopeptides. As is shown in Figure 7a, they are mainly localized on the cell, extracellular region, organelle and membrane regions in cellular components. However, upregulation was more circulated in the extracellular region and protein-containing complex region, down-regulation was also circulated in the cell junction region. As far as molecular function is concerned, it was shown that most of them participate in catalytic activity, molecular function regulator and binding, while up-regulation also played a role in transporter activity and molecular transducer activity (Figure 7b). In the aspect of biological process, both up-regulation and down-regulation are involved in the cellular process, metabolic process, cellular component organization or biogenesis, and biological regulation (Figure 7c). The effect of site-specific glycoforms on these important biological processes and molecular functions needs further investigation.

Despite the valuable results of this study, our research also has several limitations. Firstly, the HCC patient sample size was relatively small, the glycoproteins identified in urinary EVs as potential candidate biomarkers for HCC diagnosis should be further validated in larger samples. Secondly, the candidate glycoproteins were not validated by Western blot or other technologies, therefore, there is a need to evaluate the precision and robustness of the identified candidate glycoproteins as non-invasive biomarkers for HCC in any further study.

## 3. Materials and Methods

### 3.1. Patients and Sample Collection

This study was approved by the Ethics Committee of the Second Hospital of Jilin University and written informed consent was obtained from all the participants. Urine samples were obtained from 21 HCC patients (Appendix A) as well as seven normal controls. The diagnosis of HCC was determined pathologically and immunohistochemically according to the WHO classification. The first morning urine samples were collected and kept for less than 1 h at room temperature followed by centrifugation at 2000× *g* for 10 min to remove cell debris and large apoptotic bodies, and then stored at − 80 °C until use. 

### 3.2. Isolation of Extracellular Vesicles from Urine by EVTRAP

First, pooled urine samples were made by aspirating equal volumes of urine samples from HCC patients or from NC before EV isolation. Then, 10 mL of the pooled urine samples were used for EV isolation by EVTRAP method as we previously described [39]. In brief, EVTRAP beads was mixed with the urinary sample at a beads-to-urine ratio of 1:50 (*v*/*v*). The sample was incubated for 1 h with end-over-end rotation at room temperature, then a magnetic separator was used to remove the solution. Next, 150 μL of fresh 100 mM TEA solution was used twice to elute the EVs from beads, and the eluate was dried out in a vacuum centrifuge (Labconco, SpeedVac).

### 3.3. Enzyme Digestion and Protein Extraction of Urinary EVs

The dried urinary EVs were lysed to extract proteins using a phase-transfer- surfactant (PTS)-aided procedure as previously reported [63]. First, 100 μL of lysis buffer (50 mM Tris-HCl, 12 mM SDC, 12 mM SLS, 10 mM TCEP, 40 mM CAA, and 1% phosphatase inhibitor mixture in in 50 mM Tris·HCl) was used to solubilize the dried EVs. Second, the solution was boiled at 95 °C for 10 min and diluted five-fold with 50 mM TEAB. Then, the proteins were digested with Lys-C at an enzyme/proteome ratio of 1:100 (*w*/*w*) for 3 h at 37 °C. Afterwards, trypsin was added to a final 1:50 (*w*/*w*) enzyme/proteome ratio for overnight digestion at 37 °C. The digested peptides were acidified with trifluoroacetic acid to a final concentration of 0.5% trifluoroacetic acid, and 500 μL of ethyl acetate was added. After vortexing for 2 min it was centrifuged at 20,000× *g* for 2 min to obtain aqueous and organic phases. The aqueous phase was collected and desalted using the Sep-pak C18 column (Waters). Each sample was split into 5% and 95% aliquots for the proteomic and glycoproteomic experiments, respectively.

### 3.4. Stable Isotopic Dimethyl Labeling and Enrichment of N-glycopeptides from Urinary EVs

For glycoproteomic analysis, the 95% portion of each peptide sample was first labeled with isotopomeric dimethyl labels, then the N-glycopeptide enrichment was performed by the HILIC method. The stable isotope dimethyl labeling was performed as previously described with slight modifications [64]. Briefly, 200 μL of 100 mM TEAB (pH = 8.0) was first used to reconstitute the above desalted EV peptides, and 16 μL of CD2O (4%, *v*/*v*) and CH2O (4%, *v*/*v*) was added to the peptides derived from HCC and NC, respectively. After shaking for 1 min, 16 μL of 0.6 M NaBD3CN/NaBH3CN solution was added, and the solutions were incubated for 1 h in a shaker at room temperature. Next, 40 μL of 1M Tris (pH = 7.0) was used to terminate the reaction and 32 μL of formic acid was added to further quench the reaction. Afterward, the obtained light- and heavy-labeled peptides were mixed at a 1:1 ratio, desalted and dried in the SpeedVac.

The N-glycopeptides enrichment was performed with the centrifugation-assisted click maltose–HILIC approach as previously reported [65]. In brief, the dried dimethyl labeled peptide sample was reconstituted in loading buffer (80% ACN/1% TFA). Next, the peptide sample was mixed with the click maltose material in a new tube to a final volume of 200 μL and incubated for 1 h at room temperature. Then the mixture was pipetted into a 200 μL tip with a 1 mm Teflon disk blocking in the tip, and after centrifugation, the loading buffer was spun-out from HILIC tip, and the HILIC tip was washed twice with 200 μL 80% ACN/1% TFA. Finally, the enriched N-glycosylated peptides were eluted with 100 μL of 30% ACN/0.1% FA and dried with a vacuum freeze centrifuge.

### 3.5. LC−MS/MS Analysis

The dried peptides and glycopeptides were resuspended in 15 μL of 0.1% formic acid (FA) and 2% ACN solution. LC-MS/MS analysis was performed using a Dionex UltiMate 3000 RSLCnano system (Thermo Fisher Scientific, Bremen, Germany) coupled with an Orbitrap Fusion Lumos Tribrid spectrometer (Thermo Fisher Scientific, Bremen, Germany). The samples were separated on a 15 cm in-house packed column (150 μm i.d.) containing C18 AQ beads (1.9 μm, Dr. Maisch, GmbH, Germany). Mobile phase A was composed of 98% H_2_O and 2% ACN and mobile phase B was composed of 80% ACN and 20% H_2_O, both containing 0.1% FA. For the proteomic analysis, the flow rate was maintained at 600 nL/min with a linear 100 min gradient. The elution gradient was kept at 1% mobile phase B for 10 min, and then increased linearly from 1% to 34% mobile phase B over 60 min, followed by increasing linearly from 34% to 90% mobile phase B over 15 min. After remaining at 90% mobile phase B for 12 min, the content of mobile phase B was returned to 1% and maintained for 3 min. The mass spectrometer was operated in data-dependent mode. Full mass scan MS spectra (*m*/*z* 350−1800) were acquired by the Orbitrap mass analyzer with a resolution of 240,000. The AGC target for MS1 was set as 4 × 10^5^ and a maximum injection time of 50 ms. The most intense ions above a threshold ion count of 4 × 10^3^ and charge state ≥2 were selected and fragmented using HCD fragmentation with 30% normalized collision energy. The dynamic exclusion was set for 60 s with a 10 ppm mass window. Fragment ion spectra were acquired in the linear IT with an AGC of 2 × 10^4^ and a maximum injection time of 35 ms for IT MS2 detection. For the glycoproteomic analysis, the elution gradient was kept at 4% mobile phase B for 3 min, and then increased linearly from 4% to 34% mobile phase B over 60 min, followed by increasing linearly from 34% to 90% mobile phase B over 23 min. After remaining at 90% mobile phase B for 10 min, the content of mobile phase B was returned to 1% and maintained for 4 min. The flow rate was maintained at 800 nL/min. The parent ion is selected in the Orbitrap cell (FTMS) at a resolution of 120,000. Up to the top 20 most abundant isotope patterns with a charge ≥ +2 from the survey scan (350–1500 *m*/*z*) were selected with an isolation window of 1.6 *m*/*z* and fragmented by HCD with normalized collision energies of 27. The AGC target value for MS1 and MS2 scan modes was set to 5 × 10^5^ and 1 × 10^5^, respectively. System control and data collection were carried out by Xcalibur software.

### 3.6. Protein Identification

RAW files were searched against the Uniprot human database (www.uniprot.org, accessed on 20 March 2022) using MaxQuant (version 2.2) and the default search parameter settings: 10 ppm and 20 ppm mass tolerance for precursor and fragment ions, respectively; fixed modifications: carbamidomethylation of cysteine (+57.0214 Da); variable modifications: oxidation of methionine (+15.9949 Da); fully Trypsin/P digestion with up to two missed cleavages. The false discovery rates (FDR) of proteins and peptides were set at 0.01. 

### 3.7. Intact N-glycopeptideIdentification and Quantification

Database search and intact N-glycopeptides identification was performed using GPSeeker as previously described [40]. In brief, the human proteome DB (UniProt, 20,375 entries) and the human N-glycome DB (75,888 entries) were used to build the theoretical customized human intact N-glycopeptides databases [66], and the LC-MS/MS raw dataset was searched against the four databases independently. To search matching precursor and fragment ions, the isotope peak abundance cut-off, isotope peak mass-to-charge ratio (*m*/*z*) deviation and isotope peak abundance deviation were set to 40%, 20 ppm and 50%, respectively. The search of intact N-glycopeptide spectrum matches (GPSMs) included the following parameters: Y1 ion, Top5; the minimum percentage of matched fragment ions of the peptide backbone, ≥10%; the minimum matched fragment ion of the N-glycan moiety, ≥1; TopN hits, n = 2 with Top1 hit(s) having the lowest P score; G-bracket, ≥1; and GF score, ≥1. For each dataset, the GPSMs from the target and decoy searches were combined and ranked with the increasing order of the P score; a cutoff P score was chosen to obtain spectrum-level FDR ≤1%. Target GPSMs with FDR control were subjected to duplicate removal to obtain the final intact N-glycopeptide IDs. GPSeekerQuan software was used to quantify the differentially expressed intact N-glycopeptides (DEGPs) from HCC relative to NC samples. For each intact N-glycopeptide ID, GPSeekerQuan searched its paired precursor ion with a mass difference of 4.01344 Da and an isotopic peak *m*/*z* tolerance of 20 ppm. For each precursor ion, the summed abundance of Top 3 isotopic peaks was used for relative quantitation. For each intact N-glycopeptide ID, all the six isotopic peaks in the pair are required to be observed to obtain the relative ratio (HCC/NC).

### 3.8. Bioinformatics Analysis

The functional gene enrichment analysis and Vesiclepedia database search was performed with the FunRich software (version 3.1.3) [67]. Gene ontology data were analyzed using the Database for Annotation, Visualization, and Integrated Discovery (DAVID) bioinformatics tool (version 6.8, https://david.ncifcrf.gov, accessed on 20 March 2022) [68]. 

## 4. Conclusions

In this study, we performed a comprehensive glycoproteomic profiling of urinary EVs at the intact N-glycopeptide level to screen potential biomarkers for the diagnosis of HCC. Combined with DB search using GPSeeker, 756 intact N-glycopeptides with 154 N-glycosites, 158 peptide backbones, and 107 N-glycoproteins were identified. Out of 756 intact N-glycopeptides, 344 DEGPs were identified, corresponding to 308 upregulated and 36 downregulated N-glycopeptides, respectively. Compared to NC, several aberrated glycoproteins were identified in urinary EVs from patients with HCC. The glycoproteins LG3BP, PIGR and KNG1 are upregulated in HCC-derived EVs, while ASPP2 is downregulated. The findings demonstrated that specific site-specific glycoforms in these glycoproteins from urinary EVs could be potential and efficient non-invasive candidate biomarkers for HCC diagnosis.

## Figures and Tables

**Figure 1 molecules-28-01293-f001:**
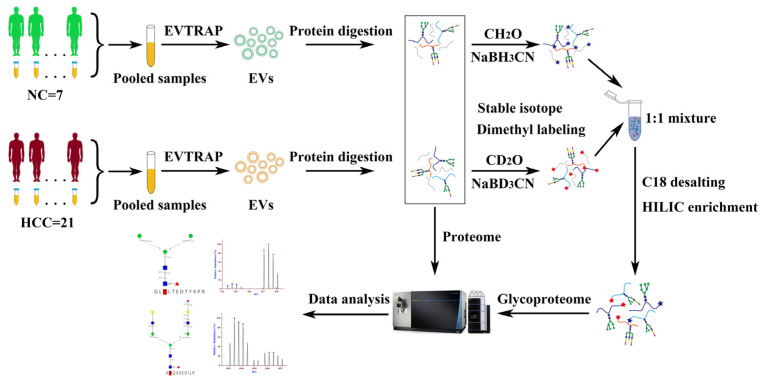
Workflow for the proteomic and glycoproteomic analysis of urinary EVs derived from patients with hepatocellular carcinoma and normal controls.

**Figure 2 molecules-28-01293-f002:**
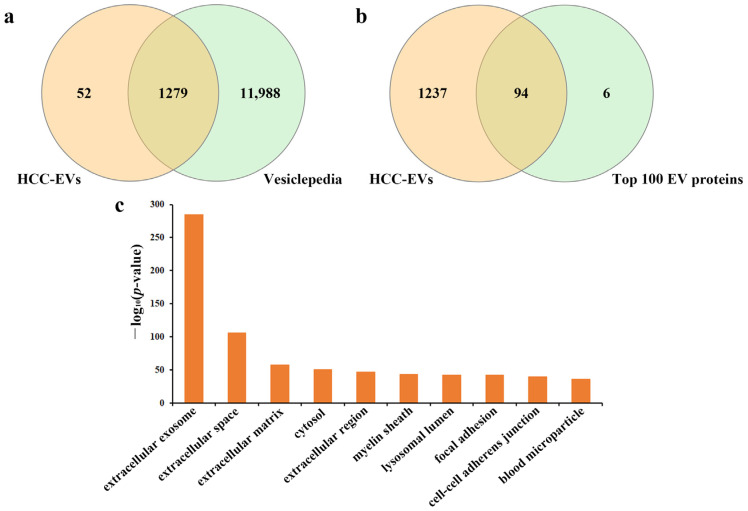
Venn diagram depicted the overlap of proteins identified from HCC-derived urinary EVs with those published in the Vesiclepedia database (**a**) and top 100 EV proteins from Vesiclepedia (**b**). (**c**) GO cellular component annotations of the identified proteins in HCC-derived urinary EVs.

**Figure 3 molecules-28-01293-f003:**
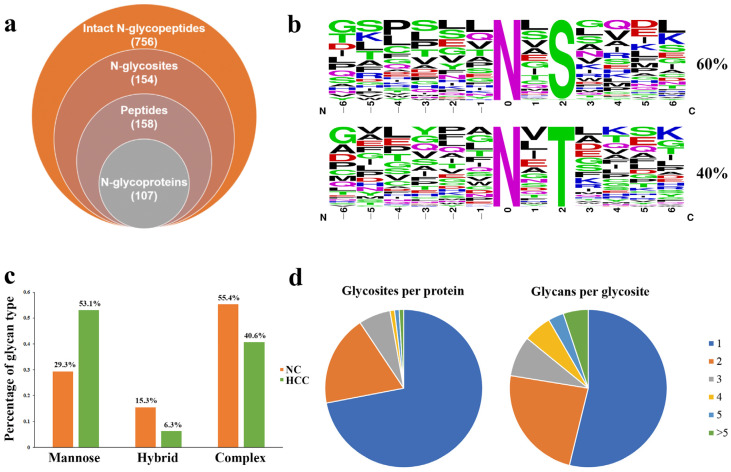
Identification and distribution of intact N-glycopeptides in urinary EVs from patients with HCC relative to NC. (**a**)The identified intact N-glycopeptide IDs (FDR ≤ 1%). (**b**) Sequence motifs for N-glycosites having either the N-XS or N-X-T sequon and their relative percentage in the unique Nglycosites identified. (**c**) The percentage of each N-glycan type in urinary EVs from patients with HCC and NC. (**d**) Distribution of glycosites (**left**) and glycans per glycosite (**right**) of the identified glycoproteins.

**Figure 4 molecules-28-01293-f004:**
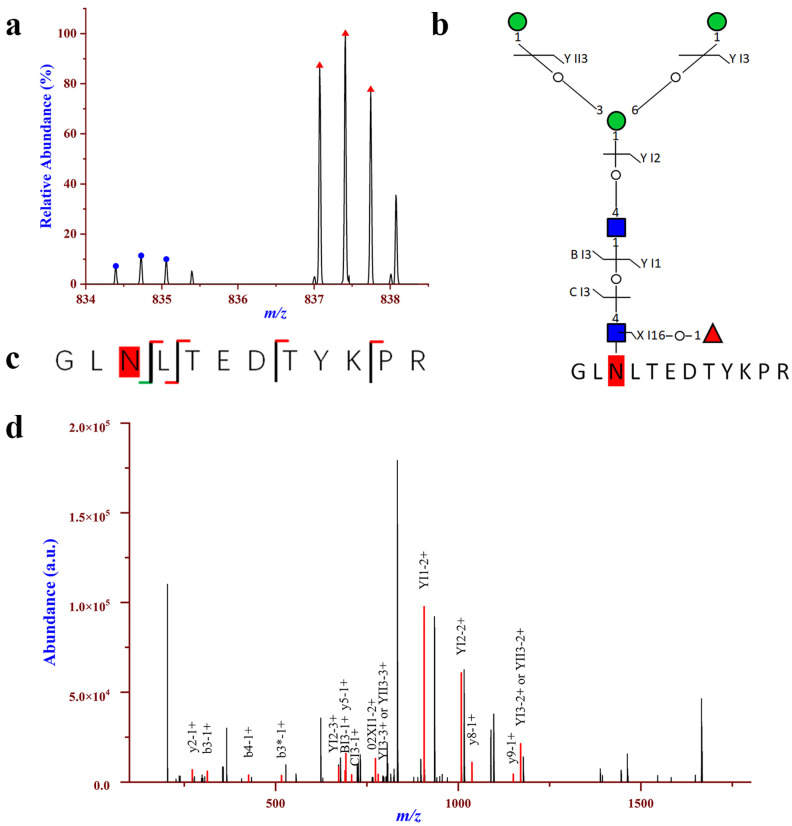
Up-regulation of intact N-glycopeptide GLNLTEDTYKPR_N2H3F1S0 in urinary EVs from patients with HCC relative to NC; the N-glycosite is N398 on Galectin-3-binding protein (LG3BP_HUMAN, Q08380). (**a**) Paired precursor ions. (**b**,**c**) N-glycan and peptide backbone graphical fragmentation maps annotated with the matched fragment ions. (**d**) The MS/MS spectrum with the matched fragment ions.

**Figure 5 molecules-28-01293-f005:**
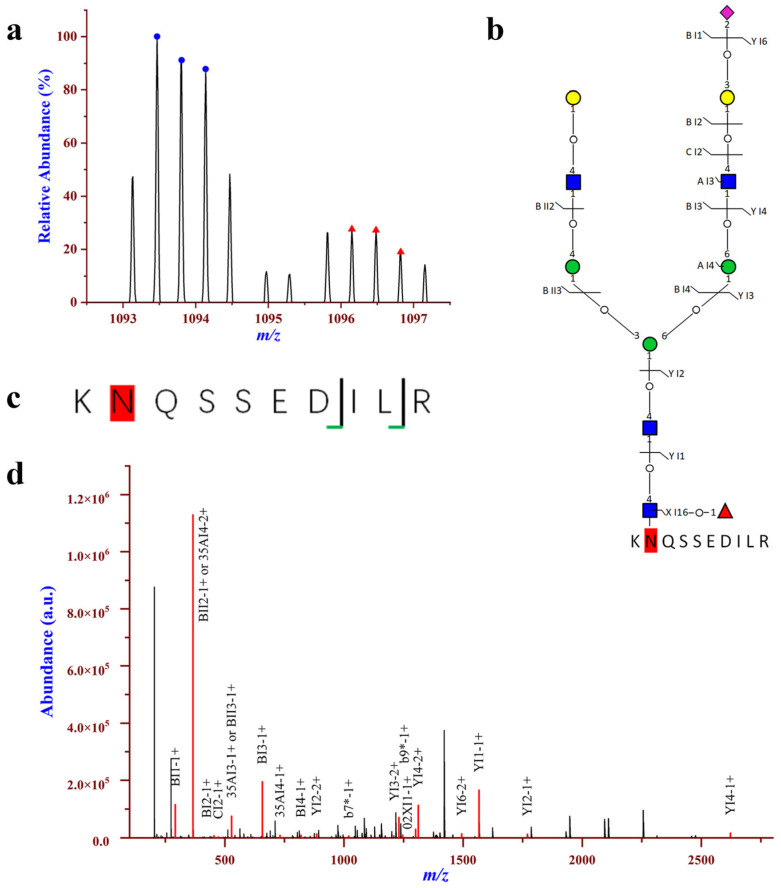
Down-regulation of intact N-glycopeptide KNQSSEDILR_N4H5F1S1 in urinary EVs from patients with HCC relative to NC; the N-glycosite is N478 on Apoptosis-stimulating of p53 protein 2 (ASPP2_HUMAN, Q13625). (**a**) Paired precursor ions. (**b**,**c**) N-glycan and peptide backbone graphical fragmentation maps annotated with the matched fragment ions. (**d**) The MS/MS spectrum with the matched fragment ions.

**Figure 6 molecules-28-01293-f006:**
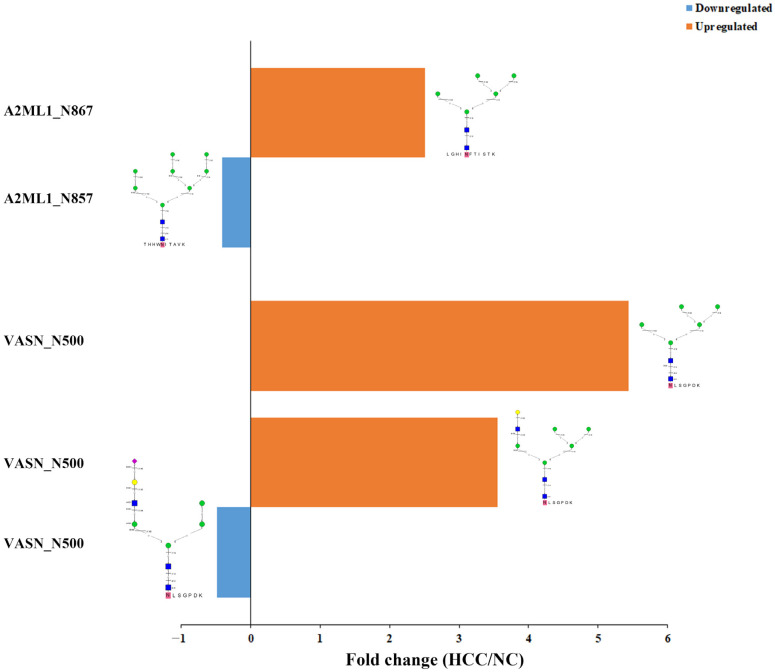
Simultaneous up- and down-regulation on alpha-2-macroglobulin-like protein 1 (A2ML1_HUMAN, A8K2U0) and vasorin (VASN_HUMAN, Q6EMK4) with different intact N-glycopeptides.

**Figure 7 molecules-28-01293-f007:**
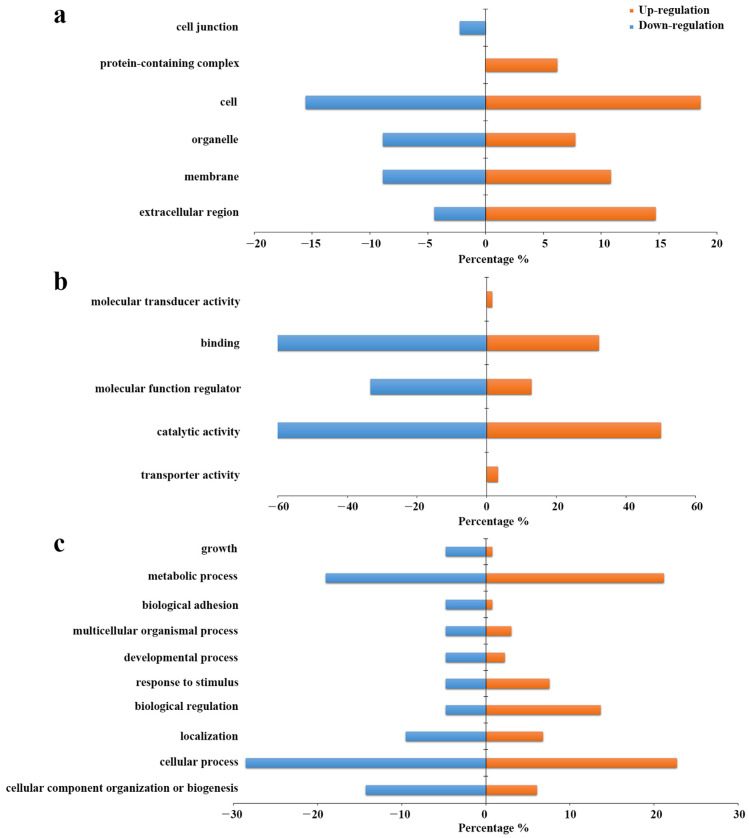
Gene ontology analysis of the intact N-glycoproteins corresponding to the differentially expressed intact N-glycopeptides in HCC relative to NC. (**a**) Cellular component. (**b**) Molecular function. (**c**) Biological process.

**Table 1 molecules-28-01293-t001:** The intact N-glycopeptides identified from probable serine carboxypeptidase CPVL (Q9H3G5, CPVL_HUMAN) modified with N2H5F0S0 on N-glycosites N81, N307 and N346 and N2H4F0S0 and N2H6F0S0 on the N-glycosite N81 and N346 as well as N2H3F0S0 and N2H4F1S0 on N-glycosite 346.

Glycosite	Peptide Sequence	Composition	Glycan-Linkage	GF Score	Structural Diagnostic Ions
81	SYAGFLTVNK	N2H4F0S0	01Y41Y41M(31M21M)61M	9	BII2,BI2,YI1,YI2,BI3,YII3,YI3,YII4,MH,YI1
		N2H5F0S0	01Y41Y41M(31M)61M(31M)61M	11	24AI4,ZI1,YI1,YI2,YI3,BI4, YI4,YII4,YIII3,MH,02XI1,ZI1,YI1
		N2H6F0S0	01Y41Y41M(31M21M)61M(31M)61M	3	BI2,YI3,YI3
307	LLDGDLTSDPSYFQNVTGCSNYYNFLR	N2H5F0S0	01Y41Y41M(31M)61M(31M)61M	3	BI2,BI4,YI1
346	QAIHVGNQTFNDGTIVEK	N2H3F0S0	01Y41Y41M(31M)61M	5	BI3,YI1,YI2,ZI1,YI1
		N2H4F0S0	01Y41Y41M(31M21M)61M	12	BII2,BI2,YI1,YI2,BI3,YII3,YI3,YII4,MH,02XI1,ZI1,YI1,YI2
		N2H4F1S0	01Y(61F)41Y41M(31M)61M61M	4	BI4,YI3,YI4,YII3,YI1
		N2H5F0S0	01Y41Y41M(31M)61M(31M)61M	12	ZI1,YI1,YI2,YI3,BI4,YI4,YII4,YIII3,02XI1,MH,ZI1,YI1,YI2,YI3
		N2H6F0S0	01Y41Y41M(31M)61M(31M21M)61M	2	BI2,YI3

## Data Availability

Not applicable.

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
