# Peer review of "Glycoproteomic Analysis of Urinary Extracellular Vesicles for Biomarkers of Hepatocellular Carcinoma"

_molecules, 2023, doi:10.3390/molecules28031293_

Round 1

Reviewer 1 Report

The manuscript by Li et al presents a thorough analysis of the glycoproteomes of extracellular vesicles (EV) in urine samples from HCC vs. normal patients.

The work is potentially impactful because liquid biopsy methods for HCC detection remain limited. The authors propose several new potential biomarkers based on differentially expressed glycoproteins in cancer vs. noncancer patients, including L3GP3 vs. ASPP2 that are upregulated vs. downregulated, respectively, in cancer patients.

The analysis is interesting because not only proteins but also glycan signatures were analyzed.

However, the work requires several points of clarification and validation to determine new biomarkers.

1. A key point of clarification is how robust the analysis is between patients. The Methods only say that “samples were pooled.” How was pooling determined? As an example, were all 21 HCC samples pooled and analyzed together, or were individual patient samples analyzed in each LC-MS/MS run? If the latter, were hits called that were observed in only 1 sample, or was there a requirement for peptides/glycopeptides to be observed in >2 samples. There is no mention of sample sizes in the paper other than 21 and 7 potentially pooled samples.

Key points: in Method section 4.1 and 4.2, Please describe how the samples were pooled. Were all 7 NC samples pooled, or all 21 HCC samples pooled? This information is critical to determine variability between individual patients or “average” EV compositions in noncancer vs. HCC. Figure S1 makes it look like 3 samples were compared, but there are no notes about how the comparison was performed.

Figure S1a: The numbers doesn’t seem to add up between a and d. Where did the total number of NC samples come from? Figure S1b: What does the term “needles” refer to when comparing overlap in the Venn diagram? Does that refer to 3 injections from the same vial, or 3 separate pools of EV samples per HCC / NC condition?

2. The main purpose of the paper is to suggest new biomarkers. Two major examples are presented with differential expression of 40-fold. For new biomarkers to be suggested based on glycan signatures and proteins, a lectin stain of the aberrant glycans in HCC vs. NC extracellular vesicle samples and/or Western blot of these proteins must be shown to confirm findings. If the results are as robust as the quantitative analysis suggests, either result would confirm the data discussed in the current manuscript. One example diagnostic protein will suffice as proof-of-concept.

3. Very little discussion is made about the overall set of differentially expressed glycoproteins. Section 2.5 runs a protein-protein interaction analysis with STRING, but I am not convinced that this is a useful analysis. What is the hypothesis for protein-protein interactions in the dataset? Also, were all DEGPs analyzed, or were just the upregulated vs. just the downregulated DEGPs analyzed? I would imagine a separate STRING analysis for up- vs. down-regulated glycoproteins would be better than all DEGPs. In any case, the meaning of the PPI analysis is not clear. Especially because the key up vs. downregulated proteins LG3BP and ASPP2 are NOT in the STRING plot, further questioning the utility of this analysis.

The GO analysis of the DEGPs in Figure S6 is useful, and in my opinion should replace Figure S7 alongside a larger discussion of the potential meanings of the results. Section 2.5 can be expanded to incorporate a more thorough analysis of the data that was collected.

4. If N-glycans are affected, O-glycans will be too. Was any analysis of O-glycoproteins performed? If so or if not, please explain further.

Minor notes:

Abstract: Please define all acronyms in the abstract.

Overall writing: The grammar needs to be verified with a spell-check system because approximately 15% of the words run together without spaces. There are far too many errors to correct by hand. Other than this issue the paper is well-written and clearly described.

Figure 2: Write “Top 100 EV proteins” as a better label for the right-hand side of panel 2b (and in the caption)- otherwise it was unclear which top 100 proteins are compared against.

Table 1: What proteins are these peptides derived from? Please add a column with Protein ID/Name. Also add the key proteins you are comparing, such as ASPP2 and LG3BP.

Figure 7: In the caption, indicate which DEGPs were compared. All up- and down-regulated? As suggested above, separate STRING analyses for up vs. down would be more helpful. I also suggested highlighting the key proteins you observed as potential biomarkers for clarity.

If these points are addressed, I support publishing in Molecules. The study is interesting and the data is potentially useful. Validation from at least one differential blot/lectin stain of a potential HCC biomarker would greatly improve the impact of the work, as will further discussion of the meaning of the dataset as a whole. I look forward to the revision.

Reviewer 2 Report

Comments to the Author

The manuscript entitled “Glycoproteomic Analysis of Urinary Extracellular Vesicles for Biomarkers of Hepatocellular Carcinoma” has been reviewed. The manuscript is organized and well written. Obviously, a lot of careful work has gone into this project. In this study, the authors found the glycoproteins LG3BP, PIGR and KNG1 are upregulated in HCC-derived EVs, while 25 ASPP2 is downregulated. The results may apply to find HCC patients from urinary EVs.

The design of this study and concept are reasonable. On the other hand, the urine was provided by 21 HCC individual. Drawback of this manuscript is that these markers were not checked against the urine of the HCC individual 21 patients from whom the urine was provided. The results of biomarkers the authors found should measure in 21 individual HCC patients. This would be most important evidence of this manuscript.  

The information of the 21 patients is limited. I think the authors need to prepare the background of patients. Do they come from HCV patients? NASH? How about progression of HCC? Sex? Age? As I spelled out above, re-check the biomarkers in urine from 21 individual patients and information of their background would be strengthen this manuscript.

Minor

Several miss typos.

Round 2

Reviewer 1 Report

The authors made the suggested changes, or else indicated in Discussion where the study is limited in terms of critical next steps. 

Author Response

Thank you very much for your valuable and constructive suggestions for the improvement of our manuscript. More details on the patients' information have been uploaded as supporting information.

Reviewer 2 Report

The manuscript entitled “Glycoproteomic Analysis of Urinary Extracellular Vesicles for Biomarkers of Hepatocellular Carcinoma” has been re-reviewed.

 I must conclude that the manuscript is incomplete and is not acceptable with this form.

The manuscript entitled “Glycoproteomic Analysis of Urinary Extracellular Vesicles for Biomarkers of Hepatocellular Carcinoma” has been re-reviewed.

It seems that the information the authors prepared as the clinical characteristics of the 21 patients in Table S1 is not fully covered, yet.

For example, the background of hepatitis and non-alcoholic steatohepatitis:(NASH)/ non-alcoholic fatty liver disease (NAFLD) for HCC is different.

Compared to hepatitis, HCC derived from NASH shows higher of AFP and lower of protein induced by Vitamin K absence or antagonist-II (PIVKA II). Besides, the incidence rate of HCC in NASH is not high. Thus, the background for HCC is extremely important. How about Child-Pugh Score? Can the authors include the score? In addition, I recommend the authors to make more detailed information for Differentiated degree in Table S1.

Again, I require the authors to validate the identified potential candidate glycoproteins in HCC patient’s samples. As the authors described, I understand the limitation of specific antibodies for glycoproteins. In addition, the authors added that the HCC patient sample size was relatively small in Discussion section.  

Even if we obtain valuable results in small number of studies, it is possible to be given unexpected results in large number of studies.

To overcome this, validation in HCC patient’s samples would be recommended. Real time PCR is impossible?

Author Response

 Thank you very much for your valuable and constructive suggestions for the improvement of our manuscript. More details on the patients' information have been uploaded as supporting information.  Real time PCR is possible to do but it can only reflect the protein abundance not the glycosylation level. We are trying to use PRM (Parallel Reaction Monitoring) to track the glycopeptide change to validate the proteomic discovery.